# A Case of Mycobacteriosis in Cultured Japanese Seabass (*Lateolabrax japonicus*) in Southern China

Zengchao Huang [1,†] , Liwen Xu [2,†], Shiping Yang [1], Shuanghu Cai [1], Jichang Jian [1] and Yucong Huang [1,*]

1. Guangdong Provincial Key Laboratory of Aquatic Animal Disease Control and Healthy Culture & Key Laboratory of Control for Diseases of Aquatic Economic Animals of Guangdong Higher Education Institutes, College of Fishery, Guangdong Ocean University, Zhanjiang 524088, China
2. Key Laboratory of South China Sea Fishery Resources Exploitation & Utilization, Ministry of Agriculture and Rural Affairs, South China Sea Fisheries Research Institute, Chinese Academy of Fishery Sciences, Guangzhou 510300, China
* Correspondence: huangyc@gdou.edu.cn
† These authors contributed equally to this work and share the first authorship.

**Abstract:** Japanese seabass (*Lateolabrax japonicus*) is an important species of cultured marine fish with high economic value in China. Nevertheless, from May to November 2019, mass mortality among cultured Japanese seabass occurred in Zhuhai City, Guangdong Province of China. Approximately 0.2–0.5% mortality was recorded daily, and the cumulative mortality was up to 30% during this disease outbreak. In this study, the clinical signs and pathological characteristics of diseased fish were investigated. Furthermore, the pathogenicity and antibiotic sensitivity of identified pathogenic bacteria from diseased fish were analyzed. The infected fish showed clinical signs of uncoordinated swimming; anorexia; pigment changes; and a number of 1–5 mm grayish-white nodules in the liver, spleen, and kidney tissues was also found. A bacterial strain, which was designated as ZHLJ2019, was isolated from the diseased fish. To ensure that ZHLJ2019 isolate was the causative agent, a Koch postulate trial was performed. Healthy Japanese seabass were infected by the intraperitoneal injection of $5 \times 10^4$, $5 \times 10^5$ and $5 \times 10^6$ CFU/fish, and cumulative mortalities within 42 days were 75%, 90%, and 100%, respectively. The bacteria colony had traditional morphological and biochemical characteristics similar to that of *Mycobacterium marinum*. Phylogenetic molecular analyses of 16S rRNA, *rpoB*, *hsp65*, *erp*, and ITS genes confirmed that the isolated strain ZHLJ2019 was *M. marinum*. The granulomatous inflammation in internal organs of Japanese seabass naturally and experimentally infected with ZHLJ2019 isolate was consistent with the classic pathological features of mycobacteriosis. Drug susceptibility of ZHLJ2019 isolate to 11 antibiotics was determined by broth dilution method in vitro. The minimum inhibitory concentrations (MICs) of minocycline, rifampicin, ethambutol, isoniazid, streptomycin, doxycycline hydrochloride, kanamycin sulfate, levofloxacin, roxithromycin, and prothionamide against the strain ZHLJ2019 were 4, 2, 8, 4, 16, 8, 8, 8, 4, and 8 μg/mL, respectively. The results of this study suggest that *M. marinum* is the causal agent responsible for the morbidity and mortality of Japanese seabass cultured in intensive brackish water dirt ponds in southern China.

**Keywords:** *Mycobacterium marinum*; Japanese seabass; Koch postulate; biochemical characteristics; granulomatous inflammation

## 1. Introduction

Mycobacteria are widely distributed in nature, especially in the aquatic environment. Piscine mycobacteriosis is one of the most common bacterial diseases caused by different nontuberculous mycobacteria (NTM), affecting natural and artificial farming populations of marine, brackish, and freshwater fish [1]. *Mycobacterium* spp. have been found to infect over 150 species of fish [2], but *M. marinum*, *M. fortuitum* and *M. chelonae* are the three mycobacterial species that dominate infected fish [3–5]. Other species, including *M. shottsii*,

*M. abscessus,* and *M. poriferae*, are also reported [6–8]. Members of the *Mycobacterium* genus are Gram-positive, nonmotile, aerobic, acid-fast, and pleomorphic organisms. The outbreaks of mycobacteriosis can cause high mortality and significant economic losses. The external clinical symptoms of fish mycobacteriosis, including pigment changes, exophthalmos, scale loss, and advanced cases of accompanied hemorrhagic lesions with penetrating muscle tissue, vary from species to species [4,9,10]. Nevertheless, the internal clinical signs of infection typically include enlargement of the liver, kidney, and spleen with grayish-white nodules (granulomas) in internal organs [1]. In addition to concern for fish health, infection of humans with NTM has been reported many times and can reach epidemic proportions [2]. *M. marinum*, *M. chelonae*, *M. fortuitum*, *M. szulgai*, and *M. gordonae* are also human pathogens and represent potential zoonosis [11–15].

Japanese seabass (*Lateolabrax japonicus*), an important marine fish species with remarkable economic value, shows a rapid growth rate and has strong adaptability to a wide range of salinity [16]. In recent years, Japanese seabass has been widely cultured in Asian countries, especially in southern China. The production of Japanese seabass in China amounted to 180,173 tons in 2019, higher than the recorded 13,592 tons in 2018 [17]. There is an increasing number of infectious diseases related to parasites and viruses due to overcrowding, poor water quality, and nutrition. These cause a serious threat to the seabass farming industry, thus causing huge economic losses to these farming countries [18,19].

Transmission of mycobacteria in fishes is believed to be mainly through ingestion of contaminated food and water [3], although vertical transmission of mycobacteria has been suggested [1]. To fulfill Koch's postulate, most experimental infections have been conducted by the intraperitoneal and intramuscular routes at present [3]. Moreover, fish can also be infected with *M. marinum* via bath exposure [20].

To date, there are no universally accepted treatments for mycobacteriosis in fish [2,9]. Most *M. marinum* infections may be acquired during the handling of the aquariums, such as cleaning or changing the water [21]. Treatment of people infected with mycobacteria often requires the use of multiple antibiotics for a long period of time [2,22]. Antibiotic susceptibility testing on fish isolates is few, and resistance appears to be dependent on both host species and pathogen strains. Rifampicin, streptomycin, and erythromycin have been shown to improve survival rates, although they are unable to fully control natural mycobacterium infections in cultured yellowtail [23]. Ethambutol, isoniazid and/or rifampicin are occasionally used to treat ornamental aquatic animals [1]. However, several isolates have also been found to be resistant to conventional anti-mycobacterial antibiotics [24].

In this study, a pathogen was isolated from the internal organs of diseased fish, and according to the results of artificial infection testing and histopathologic analysis, *M. marinum* was identified as the main pathogen causing morbidity and mortality of Japanese seabass cultured in intensive brackish water dirt ponds in southern China. *M. marinum* can cause zoonotic diseases [24]. Our analyses of the pathogenicity and antimicrobial susceptibility of *M. marinum* provide a valuable scientific reference for aquaculture treatment and public health safety.

## 2. Materials and Methods

### 2.1. Case History

A chronic disease occurred in cultured Japanese seabass reared in brackish water dirt ponds of Zhuhai City, China, from May to November 2019. The stocking density of fish in the ponds was 120,000–180,000 per hectare. Each dirt pond covers an area of 0.2–0.4 hectares with a mean water depth of 2–2.5 m and less silt at the bottom. These ponds are close to the sea, and it is very convenient to exchange with sea water. In September 2019, ten morbid Japanese seabass (body length, 25–32 cm; body weight, 350–450 g) were obtained from three ponds of a Japanese seabass aquaculture farm in Zhuhai city and immediately anesthetized with MS-222 (Sigma-Aldrich, St. Louis, MO, USA), then transported to the laboratory in large iceboxes for further investigation. Three sampling points were taken from cultural ponds to measure water physicochemical parameters at 0.5 m depth. The

parameters measured included water temperature; salinity; pH; oxygen, ammonia, and nitrite concentrations. The water temperature, salinity, pH, oxygen, ammonia nitrogen, and nitrite concentrations of the ponds were maintained at 27 °C $\pm$ 2 °C, 6‰ $\pm$ 2‰, 8.1 $\pm$ 0.1, 5.1 $\pm$ 1.5 mg/L, 0.50 $\pm$ 0.10 mg/L and 0.60 $\pm$ 0.10 mg/L, respectively (YSI 556 MPS, YSI, Inc., Yellow Springs, OH, USA).

### 2.2. Parasitological and Microbiological Examination

The gills, skin, fins, and internal organs like gall bladder, liver, spleen, heart, and kidney were detected; indeed, the examination also included the abdominal cavity and digestive tract and its contents of moribund fish were observed by light microscopy (ZEISS, DM750, Oberkochen, Germany) in accordance with standard procedures for parasite examination. The gills, skin and fins were inspected for ectoparasites. The external surfaces of all internal organs were examined for free or encapsulated parasites. The intestine was cut longitudinally and examined. The liver, spleen, kidney, heart, and muscle tissue of the body were pressed between microscopical glass sides and examined for myxozoanand protozoanby light microscopy. The nervous necrosis virus (NNV), *Megalocytivirus*, and *Ranavirus* were detected by polymerase chain reaction (PCR) or reverse transcription PCR to eliminate the presence of these viruses [25]. Samples obtained aseptically from brain, liver, spleen, and kidney tissues were directly streaked onto nutrient agar (NA); tryptone soy agar (TSA); brain heart infusion agar (BHIA) (Hopebio, Qingdao, China); Lowenstein–Jensen medium (LJM) (Hopebio); and Middlebrook 7H10 agar (Hopebio) supplemented with oleic acid-albumin-dextrose-catalase (OADC), which were later cultured aerobically at 25 °C for seven days in a biochemical incubator (Bluepard, LRH-250, Shanghai, China). After incubation, bacterial isolates were subcultured on 7H10 agar to obtain singular colonies. The purified bacterial strain designated as ZHLJ2019 was inoculated into Middlebrook 7H9 medium with OADC, stored at −80 °C, and supplemented with 15% (v/v) glycerol for further use.

### 2.3. Phenotype Testing

The culture of the isolated strain ZHLJ2019 was stained with Ziehl–Neelsen (Z&N) to observe bacterial acid-fastness. Bacterial cultures were fixed in 2.5% glutaraldehyde solution for 12 h at 4 °C; washed thrice with phosphate-buffered saline (PBS); dehydrated in ethanol solutions of 30%, 50%, 60%, 70%, 80%, 90%, and 100% for 10 min each; dried in a vacuum freeze dryer; and observed using scanning electron microscopy (SEM, ZEISS, EVO15).

Biochemical characteristics of the isolated strain ZHLJ2019, including nitrate reductase, nitrite reductase, urease, growth on 5% NaCl, Tween-80 hydrolysis, temperature tolerance, aromatic sulfatase, nicotinic acid, pyrazinamidase, thiophene-2-carboxylic acid, growth on glucose agar, growth on MacConkey agar, and pigmentation production after exposure to visible light, were also tested. The results of phenotype testing of the isolated strain ZHLJ2019 were evaluated in accordance with *Bergey's Manual of Determinative Bacteriology*. *M. marinum* ATCC 927 was used as a positive reference strain.

### 2.4. Molecular Identification

For the accurate identification of the isolated strain ZHLJ2019, molecular tools, including analyses of the 16S rRNA, RNA polymerase B subunit (*rpoB*), heat-shock protein 65 (*hsp65*), exported repeated protein (*erp*), and internal transcribed spacer (ITS) genes were applied for molecular identification. The specific primers of these genes are shown in Table 1. The genomic DNA of the isolated strain ZHLJ2019 was extracted by Tiangen Bacteria DNA Kit (Tiangen Biotech, Beijing, China) following the manufacturer's instructions. The Template DNA (100 ng) and 10 pmol of each primer were added into a PCR mixture tube containing 1× Ex Taq$^{TM}$ Buffer, 100 μM dNTPs, and 1.25 unit of Ex Taq$^{TM}$ DNA polymerase (Takara, Japan), and the volume was adjusted to 20 μL with distilled water. Table 1 also describes the PCR conditions. The amplified PCR products were subjected to agarose

gel electrophoresis separation and purified by a DNA gel extraction kit (Tiangen Biotech). The purified products were ligated into the pMD18-T vector (Takara) and sequenced by Sangon Biotech Co., Ltd. (Shanghai, China). Phylogenetic trees were constructed using the MEGA 6.0 software and the neighbor-joining method (bootstrap = 1000) on the basis of individual sequences.

**Table 1.** Primers used in this study.

| Gene | Primer Names/Sequence (5′-3′) | Product Size (bp) | References | PCR Conditions |
|---|---|---|---|---|
| 16S rRNA | 27F/AGAGTTTGATCCTGGCTCAG<br>1492R/GGTTACCTTGTTACGACTT | 1435 | [26] | 95 °C for 30 s, 51 °C for 30 s, and 72 °C for 90 s for 35 cycles |
| *rpoB* | MF/ CGACCACTTCGGCAACCG<br>MR/ TCGATCGGGCACATCCGG | 350 | [27] | 95 °C for 30 s, 61 °C for 30 s, and 72 °C for 90 s for 35 cycles |
| *hsp65* | Hsp-F/ATCGCCAAGGAGATCGAGCT<br>Hsp-R/AAGGTGCCGCGGATCTTGTT | 643 | [28] | 95 °C for 30 s, 59 °C for 30 s, and 72 °C for 90 s for 35 cycles |
| *erp* | Erp-8/GTGCCGAACCGACGCCGACG<br>Erp-9/GGCACCGGCGGCAGGTTGATCCCG | 230 | [29] | 95 °C for 30 s, 69 °C for 30 s, and 72 °C for 90 s for 35 cycles |
| ITS | MycoITS1F/GACGAAGTCGTAACAAGG<br>MycoITS1R/ATGCTCGCAACCACTATCCA | 265 | [30] | 95 °C for 30 s, 53 °C for 30 s, and 72 °C for 90 s for 35 cycles |

### 2.5. Histopathologic Examination

The organs of the infected fish, such as the liver, spleen, kidney, and heart, were fixed in a 10% neutral buffered formalin fixative. After 24 h fixation, the samples were rinsed thoroughly with PBS, transferred to 50% ethanol, and processed using industry-standard techniques. For each tissue type, 5 μm-thick sections were subjected to either hematoxylin and eosin (H&E) or Z&N acid-fast staining. The histological diagnosis of mycobacteriosis in H&E and Z&N stained sections were based on the observation of multiple granulomas and acid-fast bacilli in tissues, respectively.

### 2.6. Challenge Experiment

The Japanese seabass were used for challenge experiments to evaluate the pathogenic potential of the isolated strain ZHLJ2019. A hundred and twenty seabass (body length, 6-8 cm; body weight, 7–9 g) were purchased from a commercial farm in Zhuhai city and went through a full parasitological and microbiological examination. The fish were found clean of any disease and were designated as healthy. A total of 120 healthy Japanese seabass were randomly divided into four groups with 30 fish per group and acclimated in tanks at 25 °C for seven days before the challenge test. Each group was established with three replicates of ten fish. The isolated strain ZHLJ2019 was cultured at 25 °C, 120 rpm for a week in Middlebrook 7H9 broth supplemented with 10% albumin–dextrose-catalase (ADC) and 0.05% Tween 80 in oscillating incubator (ZHICHU, ZQTY-50N, Shanghai, China). Cultures were vortexed repeatedly with latex beads for 2 min to break up bacterial aggregates and then filtered through a 10 mm nucleopore filter to obtain a homogeneous suspension. Finally, the filtrate was then centrifuged for 10 min at low speed and diluted to the corresponding concentration with PBS (0.01 M, pH 7.2). Three groups (groups L, M and H) of Japanese seabass were injected intraperitoneally (ip) with 0.1 mL of $5 \times 10^5$, $5 \times 10^6$ and $5 \times 10^7$ CFU/mL bacterial concentrations, respectively, whereas the control group was injected ip with the same dose of PBS. Subsequently, after injection, all fish were maintained at 25 °C under the same conditions as described above. Fish were continuously recorded for morbidity and mortality and later sampled for histopathological and bacteriological examinations. The experiment ended six weeks postinoculation. Graph of cumulative percent mortality was constructed by using Graphpad Prism software version 8.0.1 (GraphPad Inc., San Diego, CA, USA).

### 2.7. Antimicrobial Susceptibility Testing

Eleven antibiotics were tested in vitro, most of which targeted for mycobacterial treatment: minocycline, rifampicin, ethambutol, isoniazid, streptomycin, doxycycline hydrochloride, kanamycin sulfate, levofloxacin, roxithromycin, prothionamide and P-

aminosalicylic acid. Tests were carried out using the broth dilution method described by Colorni [31]. Briefly, antibiotics were added into 10 mL Middlebrook 7H9 broth supplemented with 10% ADC and 0.05% Tween 80 at gradient concentrations of 0.5–64 μg/mL for each antibiotic with two-fold dilution and 1–256 μg/mL for P-aminosalicylic acid. The inoculum for the test consisted of 250 μL of 10-day-old suspension of isolated strain ZHLJ2019. The test tubes were incubated at 25 °C for 28 days. All test tube samples were plated onto TSA, and smears were stained in accordance with the Z&N technique to eliminate false-positive results caused by contaminants.

### 2.8. Statistical Analysis

The experimental results were analyzed using a one-way analysis of variance (ANOVA) in the statistical package for social sciences (SPSS v 20.0) (IBM, Armonk, NY, USA). The data were expressed as mean $\pm$ standard deviation (SD), and differences were marked by different letters (a and b).

## 3. Results

### 3.1. Disease Characterization and Clinical Signs

In September 2019, we collected diseased Japanese seabass from a farm in three mud ponds in Zhuhai City. Approximately 0.2–0.5% mortality (weight about 0.01–0.12 tons) was recorded daily, peaking from September to November. The estimated rate of mortality of the Japanese seabass in this outbreak was approximately 30% (weight of about 4.8–8 tons). The infected fish floated on the surface of the water or were on the side of the pond and showed loss of appetite, lethargy, darkened body surface, and a small number of bleeding spots on the body surface. Characteristic internal lesions were enlargement of the liver, spleen, and kidney with a remarkable number of 1–5 mm-diameter grayish-white nodules. The liver and spleen showed caseous necrosis occasionally (Figure 1). No parasites were detected in the gills, skin, fins, and internal organs of the infected fish. All tested fish were negative for NNV, *Megalocytivirus* and *Ranavirus*.

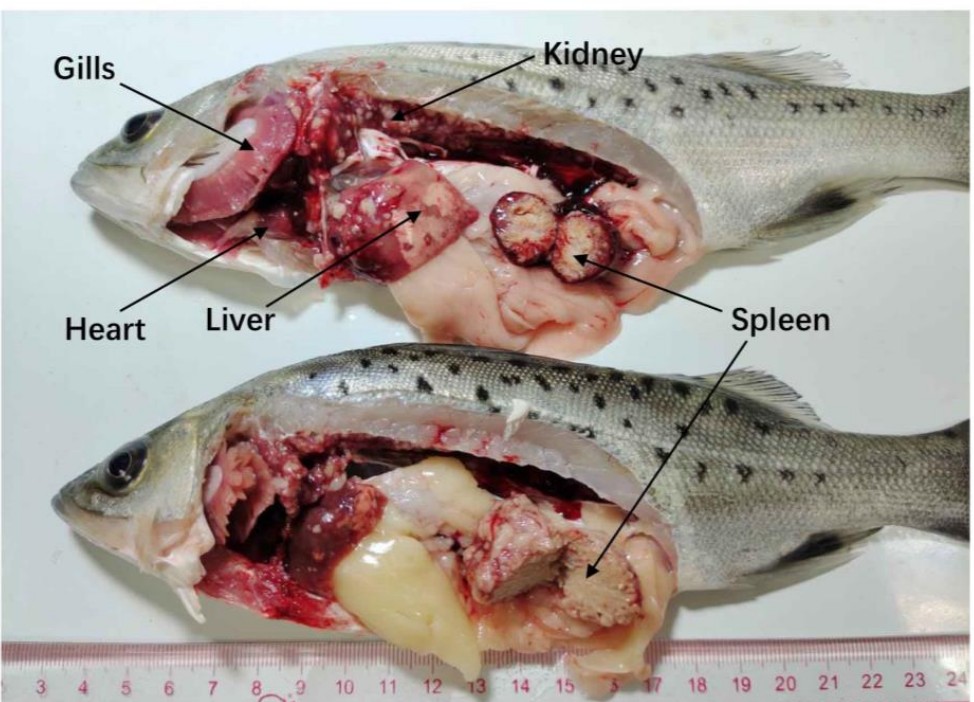

**Figure 1.** Gray-white nodules in the liver, spleen, and kidney tissues of Japanese seabass infected with *M. marinum*. Both fish were infected and their spleens were caseous necrosis.

### 3.2. Morphological and Biochemical Characteristics

The isolated strain ZHLJ2019 from the liver, spleen, and kidney of diseased fish can grow on NA, TSA, BHIA, LJM and Middlebrook 7H10 agar after three days at 25 °C and proliferated faster on BHIA and Middlebrook 7H10 agar (Figure 2A,B). Bacterial colonies were milky white in the dark and yellow to orange when exposed to visible light, smooth, and had diameters of 1.5–2.5 mm. SEM analysis showed that the isolated strain ZHLJ2019 was a slightly curved amastigote with sizes of 2–3 μm × 0.4 μm (Figure 2C) and were positive for Z&N staining (Figure 2D).

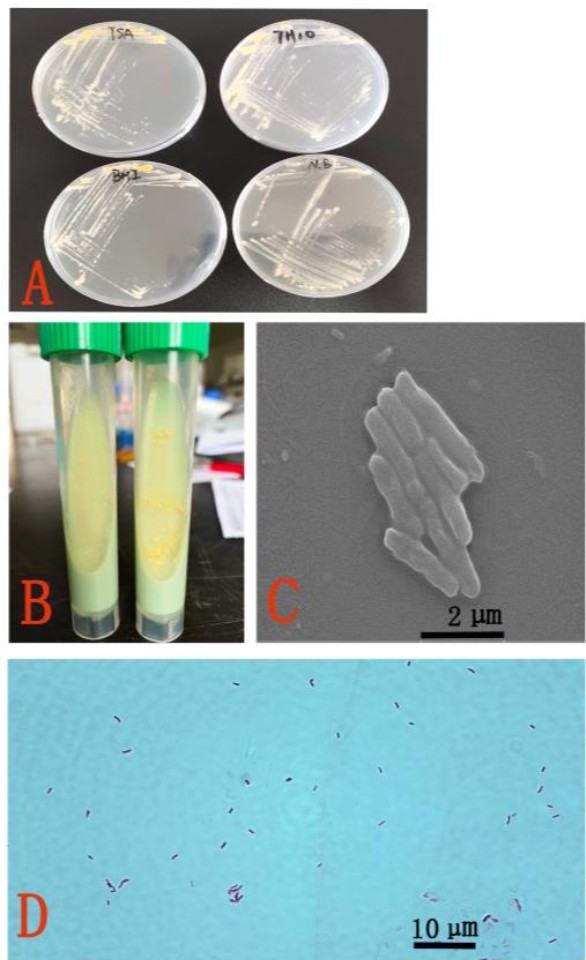

**Figure 2.** Morphological characteristics of colonies and strain ZHLJ2019. (**A**) Strain ZHLJ2019 grown on TSA, BHIA, NA, and Middlebrook 7H10 agar. (**B**) Strain ZHLJ2019 grown on LJM. (**C**) Scanning electron micrograph of the strain ZHLJ2019 (scale bar = 2 μm) (**D**). Morphological characteristics of the strain ZHLJ2019 were obtained using Ziehl–Neelsen staining (scale bar = 10 μm).

Biochemical tests of the ZHLJ2019 isolated strain showed positive results in terms of nitrite, urease, aromatic sulfatase, nicotinic acid, pyrazinamidase, Tween-80 hydrolysis, thiophene-2-carboxylic acid, glucose agar, and growth at 25 °C and 37 °C and negative results in terms of growth on 5% sodium chloride, MacConkey agar, and at 40 °C. The biochemical results of strain ZHLJ2019 were consistent with that of strain *M. marinum* ATCC927. Table 2 shows the detailed results.

**Table 2.** Comparison of phenotype characteristics of the strain ZHLJ2019 and *M. marinum* ATCC 927.

| Item | ZHLJ2019 | *M. marinum* ATCC 927 |
|:---:|:---:|:---:|
| Nitrate reduction | − | − |
| Nitrite reduction | + | + |
| Urease | + | + |
| Growth on 5% sodium chloride | − | − |
| Aromatic sulfatase | + | + |
| Nicotinic acid | + | + |
| Pyrazinamidase | + | + |
| thiophene-2-carboxylic acid | + | + |
| Glucose agar | + | + |
| MacConkey | − | − |
| 25 °C, growth | + | + |
| 37 °C, growth | + | + |
| 40 °C, growth | − | − |
| Tween-80 hydrolysis (5 day) | + | + |
| pigmentation | + | + |

*3.3. Molecular Analysis*

After PCR amplification, the fragments of 16S rRNA, *rpoB*, *hsp65*, *erp*, and ITS genes were successfully amplified from the isolated strain ZHLJ2019 and sequenced. The nucleotide sequences for these genes of the isolated strain ZHLJ2019 were submitted to GenBank under accession numbers ON108682, ON502443, ON502444, ON502445, and ON323526. The 16S rRNA gene sequence of ZHLJ2019 had high homology with mycobacteria, such as *M. marinum* strain (ATCC 927) and *M. basiliense,* with similarity exceeding 98%. In the phylogenetic tree based on 16S rRNA sequence (1435 bp) from 18 reference strains of *Mycobactrium* and one strain of *Nocardia*, the strain ZHLJ2019 was clustered with *M. marinum*, *M. ulcerans*, *M. liflandii*, and *M. pseudoshottsii* (Figure 3A). The phylogenetic tree based on the *rpoB* nucleotide sequence revealed that strain ZHLJ2019 had a close relationship with *M. marinum* (Figure 3B). The phylogenetic tree based on the *hsp65* sequence indicated that the strain ZHLJ2019 and *M. marinum*, *M. ulcerans*, *M. shottsii*, *M. liflandii*, and *M. pseudoshottsii* were in the same clade (Figure 3C). The phylogenetic analysis of the *erp* and ITS sequence revealed the clustering of strain ZHLJ2019 with *M. marinum*, *M. ulcerans*, and *M. pseudoshottsii* (Figure 3D,E).

*3.4. Histopathology*

Histopathological examination revealed that the most classic histopathological feature was granulomatous inflammation in multiple tissues, such as the liver, heart, kidney, and spleen (Figure 4A,C,E,G). Granulomas were composed of concentric layers of epithelioid cells and frequently surrounded by a fibrotic capsule. Caseous necrosis and/or calcification of the central area of granulomas were also observed in the spleen and kidney tissues (Figure 4E–G). A large number of bacilli were seen in the center of granuloma at stage II of granulomas through Z&N staining (Figure 4B–D) and in the spindle cell layers of stage III granulomas. Occasionally, adjacent granulomas appeared to fuse and form a large multinodular lesion surrounded by continuous layers of spindle and epithelioid cells (Figure 4I,J).

A (16S rRNA)

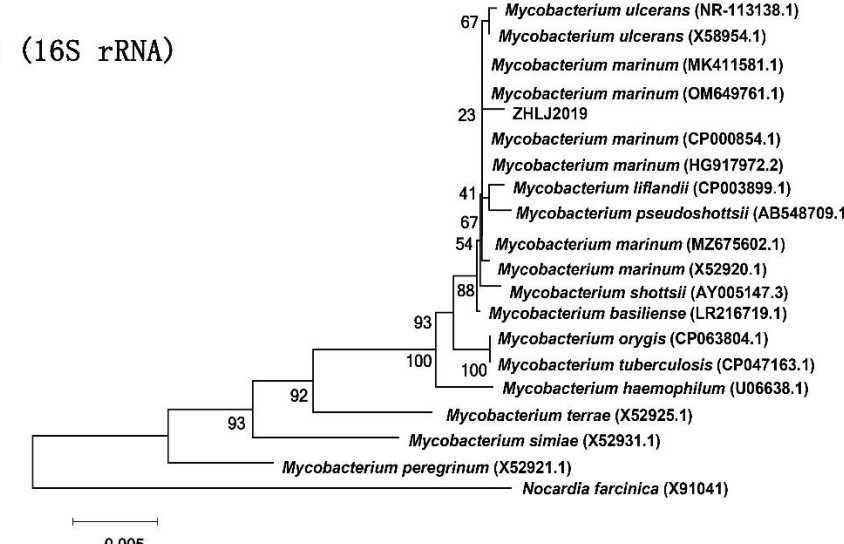

B (*rpoB*)

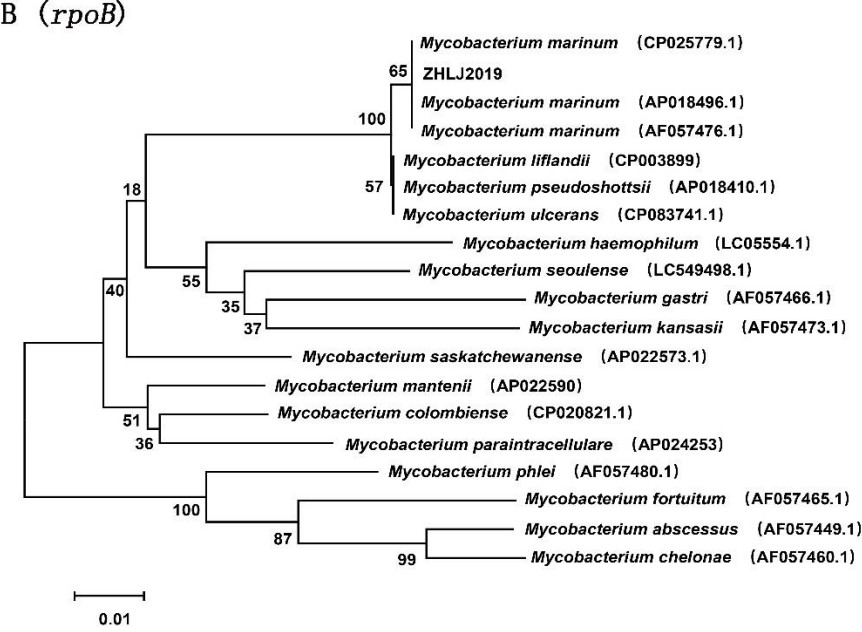

C (*hsp65*)

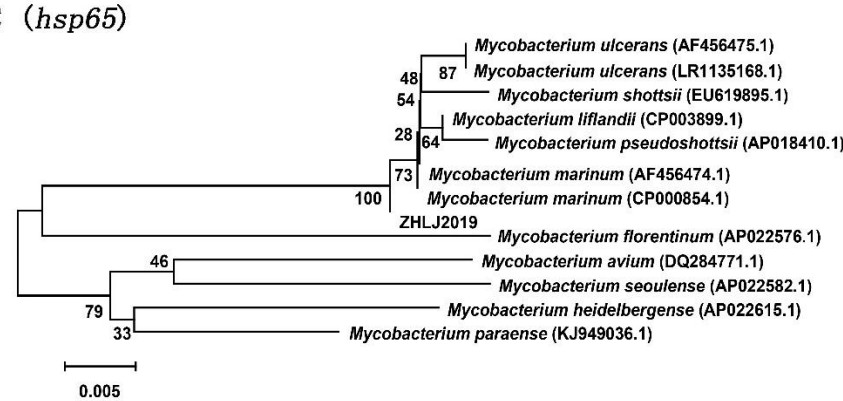

**Figure 3.** *Cont.*



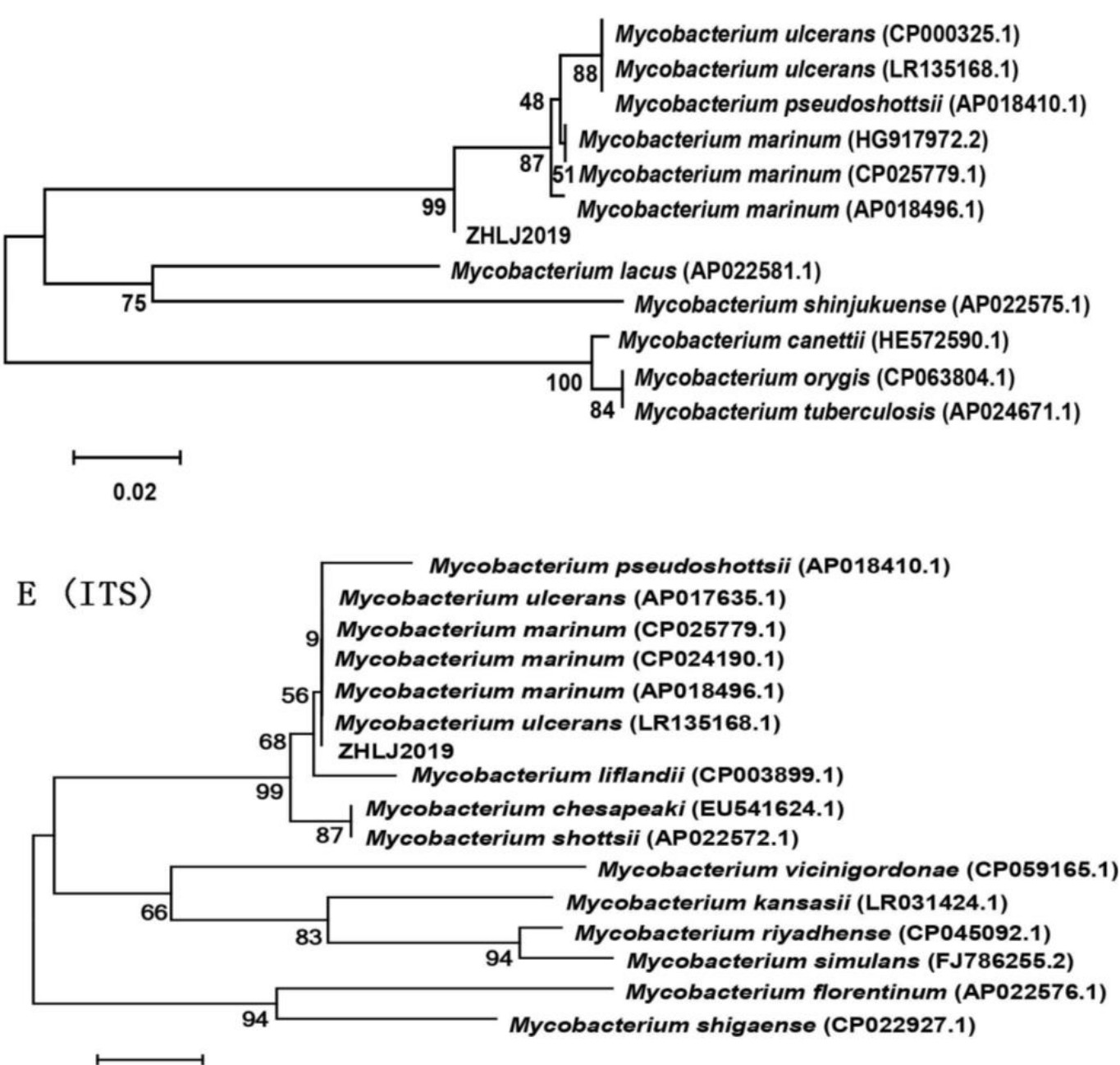

**Figure 3.** Phylogenetic trees based on the *Mycobacterium* 16S rRNA (**A**), *rpoB* (**B**), *hsp65* (**C**), *erp* (**D**), and ITS (**E**) sequences by using the neighbor-joining method. Numerical values in the tree represent bootstrap results.

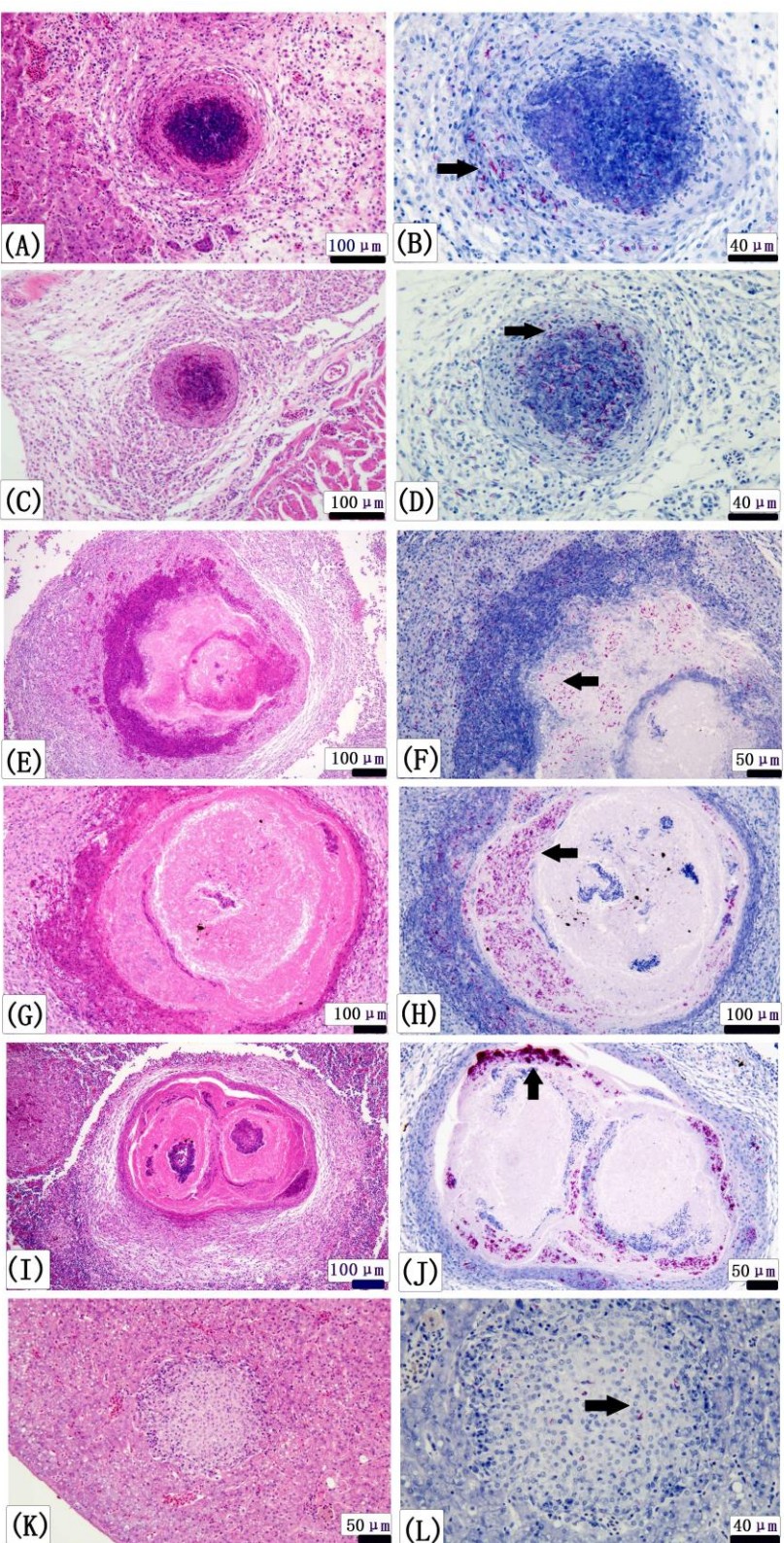

**Figure 4.** Histopathology of mycobacteriosis in Japanese seabass of natural and artificial infection. All histopathologic sections in the left column were subjected to H&E staining, and the corresponding sections in the right column were subjected to Z&N staining. (**A**,**C**,**E**,**G**,**I**) Granulomas in liver, heart, kidney, spleen, fusion of natural infection. (**K**) Spleen of artificial infection. (**B**,**D**,**F**,**H**,**J**) Numerous acid-fast bacilli in liver, spleen, kidney, heart and fusion of natural infection (arrow). (**L**) Numerous acid-fast bacilli in spleen of artificial infection (arrow).

### 3.5. Challenge Experiment

Fish mortality was first observed on day 14 post challenge. Moribund fish showed clinical signs of diffuse erythema of the skin, raised scales, and swollen abdomens. Figure 5. shows cumulative mortality rates (%, mean $\pm$ SD) of the control group, groups L ($5 \times 10^5$ CFU/mL), M ($5 \times 10^6$ CFU/mL), and H ($5 \times 10^7$ CFU/mL) were 0, 75%, 90%, and 100%, respectively, after 42 days (6 weeks). White nodules could be seen in the spleen tissue of some infected fish. The pathological features of H&E staining (Figure 4K) were similar to those of natural infection, and some acid-fast bacilli could be detected by Z&N staining (Figure 4L).

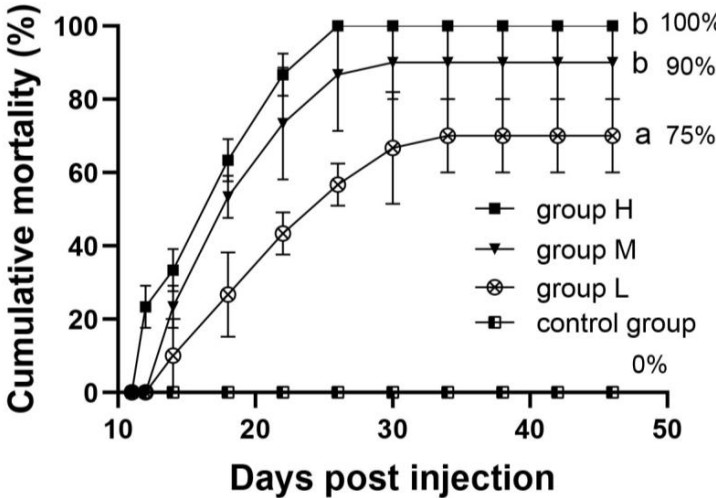

**Figure 5.** Cumulative percent mortality (%, mean $\pm$ SD) in Japanese seabass exposed to *M. marinum* and mock-injected control fish over a 6-week period. The concentrations of groups L, M, and H were $5 \times 10^5$, $5 \times 10^6$ and $5 \times 10^7$ CFU/mL, respectively. Significant differences ($p < 0.05$) are marked by different letters.

### 3.6. Antibiotic Sensitivity

The results of drug susceptibility testing are summarized in Table 3. The MIC of minocycline, rifampicin, ethambutol, isoniazid, streptomycin, doxycycline hydrochloride, kanamycin sulfate, levofloxacin, roxithromycin and prothionamide against the strain ZHLJ2019 were 4, 2, 8, 4, 16, 8, 8, 8, 4 and 8 µg/mL, respectively. The strain ZHLJ2019 was resistant to P-aminosalicylic acid in all the concentrations tested (4–256 µg/mL).

**Table 3.** Sensitivity of the strain ZHLJ2019 to conventional anti-mycobacterial drugs (+ = bacterial growth; − =growth inhibition).

| Antibiotic | Concentrations (µg/mL) | | | | | | | |
|---|---|---|---|---|---|---|---|---|
| | C | 1 | 2 | 4 | 8 | 16 | 32 | 64 |
| Minocycline | + | + | + | − | − | − | − | − |
| Ethambutol | + | + | + | + | − | − | − | − |
| Levofloxacin | + | + | + | + | − | − | − | − |
| Rifampin | + | + | − | − | − | − | − | − |
| Isoniazid | + | + | + | − | − | − | − | − |
| Roxithromycin | + | + | + | − | − | − | − | − |
| Streptomycin | + | + | + | + | + | − | − | − |
| Doxycycline hydrochloride | + | + | + | + | − | − | − | − |
| Kanamycin Sulfate | + | + | + | + | − | − | − | − |
| Prothionamide | + | + | + | + | − | − | − | − |
| | C | 4 | 8 | 16 | 32 | 64 | 128 | 256 |
| P-aminosalicylic acid | + | + | + | + | + | + | + | + |

## 4. Discussion

*M. marinum*, an opportunistic pathogen, can infect various fish from the sea and freshwater, causing morbidity and mortality in free-living fish and necrotizing granulomas similar to tuberculosis [32]. *M. marinum* is also one of the most common atypical mycobacteria, causing opportunistic infection in humans [33]. In the present study, a pathogenic *M. marinum* strain involved in a mycobacteriosis outbreak was successfully confirmed from Japanese seabass in China by bacterial isolation; morphological, biochemical, and molecular identification methods; and challenge experiment of the pathogen. To our knowledge, published reports concerning the bacterial diseases of Japanese seabass are few. This study is a new case of mycobacteriosis caused by *M. marinum* infection in farmed Japanese seabass.

In this study, specialized physiological and biochemical experiments showed that Tween-80 hydrolysis and aromatic sulfatase reaction of the isolated strain ZHLJ2019 were positive, which was consistent with those of *M. marinum* but different from those of *M. ulcerans*, *M. pseudoshottsii*, and *M. shottsii*. However, morphological traits are no longer sufficient to distinguish between closely related mycobacterial species [34,35]. Combining phenotypic tests and genotypic approaches for mycobacterial identification is necessary to prevent false-negative results.

Five moderately variable genes, i.e., 16S rRNA, *rpoB*, *hsp65*, *erp*, and ITS, were found in all mycobacterial species and chosen for phylogenetic analysis in this work. The 16S rRNA gene consisting of highly conserved domains and variable regions throughout organisms, is considered the best target to determine phylogenetic relationships and is required for mycobacterial identification [36]. The *rpoB* gene has been used for molecular identification of intestinal bacteria, rickettsia, and spirochetes, including *Borrelia* spp. [37,38], and it has been strongly recommended as a molecular tool for the identification of mycobacteria due to its good power of discrimination [27,39]. The *hsp65* gene, a single copy gene in the genome useful for differentiating the most closely related mycobacterial species or strains, is known to be more variable within the *Mycobacterium* genus than the 16S rRNA gene [28,35]. Given the large changes in sequence between *M. ulcerans*, *M. pseudoshottsii*, *M. shottsii*, and *M. marinum*, the *erp* gene is also used for species differentiation [29]. The ITS between 16S and 23S rRNA genes is considered a suitable probe target that can be used to obtain phylogenetic information [40]. On the basis of the phylogenetic tree of the 16S rRNA gene, we could infer that only the sequence of the 16S rRNA gene could not be discriminated at the species level due to high similarity among mycobacterial species. Although the *hsp65*, *erp*, and ITS genes of mycobacteria were reported to have better interspecies discrimination than the 16S rRNA gene [41], ZHLJ2019 could not be separated from other mycobacteria and still clustered with *M. ulcerans* and *M. pseudoshottsii* type strains based on the phylogenetic trees constructed by the sequences of the *hsp65*, *erp*, and ITS genes. However, the phylogenetic tree based on the *rpoB* gene indicated that the ZHLJ2019 strain was assigned to *M. marinum* and clearly differentiated from other mycobacteria. Therefore, for the identification of clinical mycobacterial strains, the sequence analysis of multiple genes, in addition to the 16S rRNA gene, should also be combined with traditional phenotypic tests to improve discrimination and accuracy. In the present study, the strain ZHLJ2019 was identified as *M. marinum* based on morphological and biochemical characteristics and phylogenetic analyses of 16S rRNA, *rpoB*, *hsp65*, *erp,* and ITS genes.

The diagnosis of fish mycobacterial disease usually includes clinical symptoms, histopathological examination, and identification of bacterial pathogens. In the present study, the diseased Japanese seabass infected by *M. marinum* showed clinical signs of enlargement of the spleen, kidney, and liver and characteristic gray or white nodules in internal organs, which were similar to that reported in half-smooth tongue sole [42], striped bass [43], yellowtail [44], and Atlantic cod [45]. In addition, the histology of diseased fish analysis revealed different-sized granulomas at various developmental stages and the presence of acid-fast bacilli within granulomas in the liver, spleen, and kidney tissues, which matches the feature of *Mycobacterium* species.

Granulomatous inflammation is the typical histopathological manifestation of fish mycobacteriosis [1]. Granuloma, as a well-defined nodular lesion formed by local infiltration and proliferation of macrophages and their evolved cells, regularly results from microbial infections, inflammation, or the presence of foreign substances and is a strategy of the immune system to isolate microbes or substances that cannot be eliminated [46]. Fish infected with *M. marinum* usually undergo five morphologically distinct stages of a temporal progression: inflammatory focus (−), epithelioid granuloma (stage I), spindle cell granuloma (stage II), bacillary granuloma (stage III), and recrudescent lesion (stage IV) [3]. In the present study, granulomas were well-organized and composed of multiple layers of epithelioid cells and proliferating fibroblasts, with necrosis in the central area of late lesions. Similar findings were found in striped bass [47], half-smooth tongue sole [40], hybrid striped bass [44], Atlantic cod [45], and freshwater ornamental fish infected with mycobacteria [48]. The structure of granulomas in farmed pikeperch is poorly formed with a lack of fibers [49], representing a relatively early stage of granuloma development. Meanwhile, fish granuloma is caused by a variety of pathogens, and the morphological differences between different fish species and different pathogens are not clear. Thus, the structure of granulomas is closely related to the causative agent, host species, nutritional factors [43,50], and their maturity [48]. The morphological variation of nodular lesions associated with different fish spp., organs and different *Mycobacterium* spp. needs to be further studied.

Antibiotic susceptibility testing on mycobacteria is rarely performed. The susceptibility pattern of *M. marinum* towards antituberculous drugs was determined in the present study. Results showed that rifampicin, isoniazid, and roxithromycin exhibited antibacterial activity against *M. marinum*, which was similar to that described in seabass [31]. However, given its low MIC for *M. marinum* in vitro, rifampicin has been proven ineffective in eradicating the disease in vivo [31]. Likewise, success has not been achieved in combating fish mycobacteriosis with other antibiotics [23,51]. We speculate that this is related to the antibiotic resistance of mycobacteria. Antibiotic resistance is mainly due to chromosomal mutations in genes encoding its target or activating enzymes, and they can be spontaneous or drug-induced after receiving [52,53]. In humans, *M. marinum* infection is usually confined to the skin, but it can spread to deeper structures, causing tenosynovitis, arthritis, and osteomyelitis [54,55]. Humans may be infected by direct injury from the fish fins or bites, but there is a lack of drug sensitivity data (MIC) on *M. marinum* derived from diseased fish [56–58]. Therefore, the drug sensitivity test can provide a reference for the clinical medication of *M. marinum* derived from fish harbored mycobacteria.

At present, no effective treatment for mycobacteriosis in fish is available except for the timely removal of diseased and dead fish after *Mycobacterium* infection [24]. However, due to the slow progression of fish mycobacteriosis and the lack of evident clinical signs, mycobacteriosis is not easy to detect, and taking isolation measures in the early stage of the disease is difficult. In addition, although the specific transmission route is not clear, many studies showed that mycobacteria can spread horizontally through aquaculture water, aquaculture equipment, and food [22]. The prevention strategy for *M. marinum* is strict disinfection of aquaculture water and facilities, and quarantine measures should be taken to prevent mycobacteria from entering the aquaculture pond. The epidemiological investigation of mycobacteriosis in fish is a priority task for public health. Furthermore, the development of effective vaccines is extremely urgent to protect fish and humans from *M. marinum* infection.

## 5. Conclusions

This is an important case of mycobacteriosis among Japanese seabass cultured in China. The causative agent of this case has been identified as *M. marinum* based on bacteriological, histopathological and molecular analysis. Rifampicin, isoniazid and roxithromycin have antibacterial activity against the *M. marinum* strain. The results will lay the foundation for

further research on diagnostic technology, virulence mechanisms, and vaccine development for the prevention and control of this disease in aquaculture and public health.

**Author Contributions:** Data curation, Z.H. and S.Y.; Formal analysis, L.X. and J.J.; Project administration, J.J. and Y.H.; Resources, L.X.; Software, S.Y.; Supervision, S.C.; Validation, Z.H. and Y.H.; Visualization, S.C.; Writing—original draft, Z.H.; Writing—review & editing, Z.H. and L.X. All authors have read and agreed to the published version of the manuscript.

**Funding:** This research was funded by the Zhanjiang Science and Technology Plan Project (2021A05196), National Key R & D Program of China (2022YFD2401200) and the Key-Area Research and Development Program of Guangdong Province (2021B0202040002).

**Institutional Review Board Statement:** All animal experiments were conducted strictly based on the recommendations in the '*Guide for the Care and Use of Laboratory Animals*' set by the National Institutes of Health. The animal experiments were approved by the Animal Ethics Committee of Guangdong Ocean University (Zhanjiang, China), approval code: (2022)1, approval date: 10 April 2022.

**Informed Consent Statement:** Not applicable.

**Data Availability Statement:** All raw sequencing data have been deposited in NCBI, and accession numbers were ON108682, ON502443, ON502444, ON502445, and ON323526 (https://www.ncbi.nlm.nih.gov/).

**Acknowledgments:** We thank Carola Wang for editing the English text of various drafts of this manuscript.

**Conflicts of Interest:** The authors declare no conflict of interest.

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
