# Peer review of "A Case of Mycobacteriosis in Cultured Japanese Seabass (Lateolabrax japonicus) in Southern China"

_fishes, doi:10.3390/fishes8010033_

Round 1
Reviewer 1 Report
In this article, Huang et al. present a case of an outbreak of mycobacteriosis in Japanese seabass (Lateolabrax japonicus) cultured in brackish water in Southern China. To identify the etiological agent of the mass mortality recorded in 2019, the authors characterized the pathology of the disease, isolated ZHLJ2019 from infected tissues, and characterized the morphology and biochemistry of the isolate. Ultimately, ZHLJ2019 was identified as the species Mycobacterium marinum through phylogenetic analyses of several genes that are capable of distinguishing between highly related Mycobacterium species. I was particularly delighted by how this is a modern example of testing if an isolated bacterium follows Koch's postulates.
Overall, the study is comprehensive, vigorous, well written, and certainly merits publication for its implications for fish welfare, fish farming, the prevalence of piscine mycobacteriosis and specifically M. marinum that make it likely such an outbreak will occur again in the near future. Furthermore, it is relevant for basic research for scientists interested in the immunological phenomenon of granulomas (in teleost fishes). Any suggestions I have are minor and thus do not require major revisions to address.
Although I am very satisfied with the editing, please correct these few lingering mistakes:
Line 47 - Please change "epidemic proportion" into the plural form "epidemic proportions".
Line 143 - There is redundancy here. Please change to either "six weeks postinoculation" or "six weeks after inoculation".
Line 150 - If I understand correctly, 10 mL of antibiotics-containing broth was used to test each antibiotic. What does the size of the inoculum refer to then, the "250 mL"? Was it 250 µL that was inoculated into each 10 mL of broth or does the "250 mL" refer only to the size of the 10-day-old suspension? If it is the latter, it does not seem like relevant information.
Line 186 to 188 - Please correct the mislabeling of Figure 2. C) should refer to the "Scanning electron micrograph of the strain ZHLJ2019..." whereas D) should refer to the Z&N staining. Not the other way around.
Line 248 - Please remove the extra period after "Mycobacterium" or use the abbreviation "M. marinum" as it is not the first instance of the bacterium being mentioned.
Line 251 - Please correct the typographical error: "opportunistic" instead of "opportunisitic".
Line 252 - Please add the article "a" before "mycobacteriosis outbreak": "... involved in a mycobacteriorsis outbreak..."
Line 258 - Please change "Specialized" to lowercase: "... specialized physiological and biochemical..."
Line 301 - Do the authors mean "... which matches the features of infections by Mycobacterium species."?
Content-wise, these are suggestions that the authors can consider and change/add at their discretion:
Section 3.5 - ZHLJ2019 follows almost all of Koch's postulates with the exception of it being reisolated from animals in the challenge experiment, to prove it is the etiological agent. This is not necessary because the control group, the infectivity, and the virulence of the isolate are sufficient to prove that ZHLJ2019 is the cause of the disease. However, especially since the authors have done it as explained on line 142, can the authors present the results of the "histopathological examinations" as an indirect way to prove that the same disease was reproduced? Section 3.5 only refers to the mortality of the challenged fish. Especially because it was a large experiment with 120 fish, maybe more of the results can be presented to make the experiment worthwhile. Did the challenged fish also have "gray-white" nodules in infected tissues?
On an unrelated note about the challenge experiment, the bacterium seems extremely virulent with even the smallest dose killing the majority of the fish. Do the authors have any idea how few bacteria are necessary for a productive infection? This could be relevant to understanding how susceptible fish are in fish farms and what requirements there are for an outbreak. Is this a vast excess relative to the size of the fish? The fish were 7 cm on average. About what mass are these fish (e.g., in grams) so that others may reproduce these results if desired, in this species of fish as well as others?
Finally, is there anything know about antibiotic resistance to Mycobacterium species? Is that also an explanation as to why it is not an effective form of treatment? This may be worth discussing if the authors agree that it is a relevant topic for aquaculture, pathogen control, and fish farming. Furthermore, there is a lot of attention directed to antibiotic resistance of bacteria species that infect humans. Especially since M. marinum is an opportunistic human pathogen, discussing how antibiotic treatment is or is not feasible could strengthen the Discussion section and makes the authors' antibiotic sensitivity experiments much more valuable.
Author Response
Dear Editor and Reviewers:
Thank you very much for your letter and the comments from the reviewers about our paper submitted to Fishes (Diagnoise and Treatment of Disease in Fish). The manuscript was carefully revised by using red text according to the comments. We responded point by point to each reviewer’s comments as listed below.
Changes for review issues are shown in blue. We supplemented some results of experiment. Histological sections of visceral organs of Japanese seabass experimentally infected with the isolated strain of ZHLJ2019, were list as Fig (4K-L). Figure 5 was remade.
Hope these will make it more acceptable for publication.
Thank you very much again.
Sincerely,
Zengchao, Huang
Point 1: Line 47 - Please change "epidemic proportion" into the plural form "epidemic proportions
Response 1: Thanks for your suggestion. I have changed "epidemic proportion" into the plural form "epidemic proportions" in the manuscript. (line 54)
Point 2: Line 143 - There is redundancy here. Please change to either "six weeks postinoculation" or "six weeks after inoculation"
Response 2: Thanks for your suggestion. I have changed to "six weeks postinoculation”(line 180).
Point 3: Line 150 - If I understand correctly, 10 mL of antibiotics-containing broth was used to test each antibiotic. What does the size of the inoculum refer to then, the "250 mL"? Was it 250 µL that was inoculated into each 10 mL of broth or does the "250 mL" refer only to the size of the 10-day-old suspension? If it is the latter, it does not seem like relevant information
Response 3: Thanks for your guidance. We are so sorry to make the mistake. I have changed "250 mL " into "250 µL ".(line 187)
Point 4: Line 186 to 188 - Please correct the mislabeling of Figure 2. C) should refer to the "Scanning electron micrograph of the strain ZHLJ2019..." whereas D) should refer to the Z&N staining. Not the other way around.
Response 4: We are so sorry to make the mistakes. We have changed into "(C) Scanning electron micrograph of the strain ZHLJ2019 (scale bar = 2 μm) (D). Morphological characteristics of the strain ZHLJ2019 obtained using Ziehl–Neelsen staining (scale bar = 10 μm).in the manuscript".(line23-225)
Point 5: Line 248 - Please remove the extra period after "Mycobacterium" or use the abbreviation "M. marinum" as it is not the first instance of the bacterium being mentioned
Response 5: Thanks for your suggestion. I have changed " Mycobacterium marinum " into " M. marinum". in the manuscript.(line 292)
Point 6: Line 251 - Please correct the typographical error: "opportunistic" instead of "opportunisitic"
Response 6: Thanks for your suggestion. I have changed " opportunistic " into "opportunisitic "in the manuscript.(line 295)
Point 7: Line 252 - Please add the article "a" before "mycobacteriosis outbreak": "... involved in a mycobacteriorsis outbreak..."
Response 7: Thanks for your suggestion. I have added "a" before "mycobacteriosis outbreak" in the manuscript(line 296).
Point 8: Line 258 - Please change "Specialized" to lowercase: "... specialized physiological and biochemical..."
Response 8: Thanks for your suggestion. I have changed "Specialized" into "specialized " in the manuscript. (line 301)
Point 9: Line 301 - Do the authors mean "... which matches the features of infections by Mycobacterium species."?
Response 9: Thanks for your suggestion. I have changed "conformed" into "matches".(line 344)
Point 10: ZHLJ2019 follows almost all of Koch's postulates with the exception of it being reisolated from animals in the challenge experiment, to prove it is the etiological agent. This is not necessary because the control group, the infectivity, and the virulence of the isolate are sufficient to prove that ZHLJ2019 is the cause of the disease. However, especially since the authors have done it as explained on line 142, can the authors present the results of the "histopathological examinations" as an indirect way to prove that the same disease was reproduced? Section 3.5 only refers to the mortality of the challenged fish. Especially because it was a large experiment with 120 fish, maybe more of the results can be presented to make the experiment worthwhile. Did the challenged fish also have "gray-white" nodules in infected tissues?
Response 10: Thanks for your suggestion. I have added the results of histopathological examinations in the manuscript (Fig 4K-L). Gray-white" nodules were found in the splee of some challenged fish.
Point 11: On an unrelated note about the challenge experiment, the bacterium seems extremely virulent with even the smallest dose killing the majority of the fish. Do the authors have any idea how few bacteria are necessary for a productive infection? This could be relevant to understanding how susceptible fish are in fish farms and what requirements there are for an outbreak. Is this a vast excess relative to the size of the fish? The fish were 7 cm on average. About what mass are these fish (e.g., in grams) so that others may reproduce these results if desired, in this species of fish as well as others?
Response 11: Thanks for your suggestion. In this study, we only make an investigation in M. marinum in the Janpanese seabass under experimental conditions, the dose of productive infection needs to further studied. In this stuy, the weight of Japanese seabass (body length, 6-8 cm; body weight, 7-9 g) have been added to the manuscript.
Point 12: Finally, is there anything know about antibiotic resistance to Mycobacterium species? Is that also an explanation as to why it is not an effective form of treatment? This may be worth discussing if the authors agree that it is a relevant topic for aquaculture, pathogen control, and fish farming. Furthermore, there is a lot of attention directed to antibiotic resistance of bacteria species that infect humans. Especially since M. marinum is an opportunistic human pathogen, discussing how antibiotic treatment is or is not feasible could strengthen the Discussion section and makes the authors' antibiotic sensitivity experiments much more valuable
Response 11: Thanks for your suggestion. “Antibiotic resistance mainly due to chromosomal mutations in genes encoding its targe or activating enzymes, and they can be spontaneous or drug-induced after receiving[47-48]. In humans, M. marinum infection is usually confined to the skin, but it can spread to deeper structures, causing tenosynovitis, arthritis, and osteomyelitis[49-50]. Humans may be infected when dealing with diseased fish, but there is a lack of drug sensitivity data (MIC) on M. marinum derived from diseased fish. Therefore, the drug sensitivity test can provide a reference for the clinical medication of M.marinum derived from diseased fish to infect people.”

Reviewer 2 Report
Added as a word document

Author Response
Dear Editor and Reviewers:
Thank you very much for your letter and the comments from the reviewers about our paper submitted to Fishes (Diagnoise and Treatment of Disease in Fish). The manuscript was carefully revised by using red text according to the comments. We responded point by point to each reviewer’s comments as listed below.
Changes for review issues are shown in blue. We supplemented some results of experiment. Histological sections of visceral organs of Japanese seabass experimentally infected with the isolated strain of ZHLJ2019, were list as Fig (4K-L). Figure5 was remade.
Hope these will make it more acceptable for publication.
Thank you very much again.
Sincerely,
Zengchao, Huang
Point 1: Abstract: This part should be rewritten it does not summarize in a clear the rezone, method and what was discovered in this study.
Response 1: Thanks for your advices. I have rewrote this part, specifiy in the manuscript.(line13-34)
Point 2: Line 15- among cultured Japanese-extra space between the two bold words
Response 2: Thanks for your advices. I have deleted extra space among cultured Japanese. (line 15)
Point 3: Line 16- Approximately 0.2%–0.5% mortality is recorded daily, and the cumulative mortality
is up to 30% during this disease outbreak. - Change to was
Response 3: We are so sorry to make the mistakes. I have change ‘Is’ to ‘was’. (line 16)
Point 4: Line 10-21- The bacterial strain designated as ZHLJ2019 was isolated from diseased fish and had the traditional morphological and biochemical characteristics of Mycobacterium marinum. Phylogenetic analyses of 16S rRNA, rpoB, hsp65, erp, and ITS genes showed that the isolated strain ZHLJ2019 was M. marinum. Change:
Response 4: Thanks for your advices. I have change this sentence in the manuscript. (line 20-25)
Point 5: Line 23-25-This part is out of context, there is a need to clarify if this was a controlled infection, a Koch postulate, what was this injection of bacterial isolate done for? and why?
Response 5: Thanks for your guaidance. I have describe as “To follow Koch postulate, healthy Japanese seabass were infected by the intraperitoneal injection of 5 × 104, 5 × 105, and 5 × 106 CFU/fish, and cumulative mortalities were 75%, 90%, and 100%, respectively, within 42 days.” (line 21-23)
Point 6: Line 26-27-Japanese seabass cultured in intensive brackish water in southern China. What culture system inland or sea cages. Origin of fish? Mycobacteria affects many spp. of fish reared in different salinity's. It can break out in any facility. What is unique to this intensive brackish water culture facility that makes it stand out?
Response 6: Thanks for your advices. Janpanes seabass is the second largest production of marine fish in China. There are three mode of culture system: salinization culture of dirt pond, salinization culture of marine cage and salinization culture of cement pond. In this study, the culture system had outbreak diseased was intensive brackish water dirt pond and fish was original commercial fish hatchery. This is an earlier case of M. marinum infected Janpanese seabass cultured in the intensive brackish water dirt pond in southern China.
Point 7: I would be cautious in stating a "This study is the first report of M. marinum infection" Mycobacterium gordonae was isolated from 24 samples collected from the fish. Mycobacterium abscessus was isolated from 3 fish samples (Lateolabrax japonicus 1 and Sciaenops ocellatus 2). M. abscessus and M. gordonae were isolated from all water samples. This investigation provides strong evidence that the predisposing factor for the M. marinum infection was with a fish spine injury acquired at a gambling fishing pond. The source of the infection was the contaminated pond water. ( H.-C. Tsai et al. 2007; Diagnostic Microbiology and Infectious Disease 59; 227–230
Response 7: Thanks for your warm comment. M. marinum is an opportunistic pathogen, can infect various fish, Zhang Luo et al was reported M. marinum was infected half-smooth tongue sole (Cynoglossus semilaevis Günther) in China; E Bozzetta et al were reported M.marinum infected in a hybrid striped bass farm in Italy. ( Hashish, E. et al. 2018; Veterinary Quarterly, 38, 35-46.; Luo Z et al. 2018; Aquacuture 490, 203-207; E Bozzetta et al.2010; Journal of Fish Disease 33, 781–785 ). In this study, M. marinum is the causal agent responsible for the morbidity and mortality of Japanese seabass cultured in intensive brackish water dirt ponds in southern China.
Point 8: Line 54-56- Change to- There's an increasing number of infectious diseases outbakes related
to parasites and virus reported due to overcrowding, poor water quality, and nutrition. These
cause a serious threat to the seabass farming industry, thus causing huge economic losses to
these farming countries [18].
Response 8: Thank you for your valuable advice. I have changed “The” to ” There’s an”, ” thus causing” to ” These cause”.
Point 9: Line 60-61- what type of culture facility need more description.
Response 9: Thank you your valuable advice. I have have revise the sentence as “M.marinum was identified as the main pathogen causing morbidity and mortality of Japanese seabass cultured in intensive brackish water dirt ponds in southern China.”
Point 10: Line 63- antimicrobial drug test of M. marinum are described.-Not described in the abstract
also, not described or explained in the introduction why these antimicrobials were chosen or
why the author decided to run the tests. Context is missing.
Add paragraphs to give context by describing why a challenge was done? and why the drug
sensitivity was done? and give references.
Response 9: Thank you your valuable advice. I have added two paragraphs in the introduction.(line65-78)
Point 11: Line 66- in brackish water ponds- Please give a more descriptive outlook of the ponds this is the first time in the manuscript it says ponds. But, are they concrete or dirt ponds what is the
water sores? Is it a flow through system?
Response 11: Thank you your advices. “Each dirt pond covers an area of 0.2 - 0.4 hectare, the depth of water is 2 - 2.5 meters, less silt at the bottom of the pond and is close to the sea to facillitate the trasportation of seawater.”
Point 12: Line 70-71- correct to- The parameters measured included water temperature; salinity; pH;
oxygen, ammonia and nitrite concentrations.
Response 12: Thanks for your advice. I have corrected it in the manuscript.(line 98-99)
Point 13: Where the fish obtained for diagnosis from three different ponds and where there differences in the fish size or age? How many were obtained from each pond? Where the fish just put on ICE or where they euthanized before and if so, what medication was used?
Response 13: Thanks for your advices. “In Septemtber 2019, ten morbid Japanese seabass (body length, 25-32 cm; body weight, 350-450g ) were obtained from three ponds from a Janpanese seabass aquaculture plant in the Zhuhai city and immediately anesthetization with MS-222 (sigma-Aldrich, St. louis, MO, USA), then transported to the laboratory in large iceboxes for further investigation.”
Point 14: Line 78- VNN, megalocytivirus, and ranavirus were detected by PCR or RT-PCR
Where the fish positive to those viruses? Or where the tissues analyzed to eliminate the
presence of these viruses by PCR or RT-PCR?
Response 14: Thanks for your advices. These fish from the fish sampled in this study. “The nervous necrosis virus (NNV), megalocytivirus, and ranavirus were detected by PCR or RT-PCR to eliminate the presence of these viruses [22]”
Point 15: In General, the materials and methods part of the Parasitological and microbiological
examination is missing information such as the companies the different agars that were used
were purchased from or if they were made the protocols. The incubation temperatures for the
different agar incubator used with company and manufacturer names.
Response 15: Thanks for your suggestions. “The intestine was cut longitudinally and examined. The liver, spleen, kidney, heart, and muscle tissue of the body are pressed between microscopical glass sides and examined for parasites by light microscopy.” “(hopebio, Qingdao, China).” “ in biochemical incubator (bluepard, LRH-250, Shanghai, China)”
Point 16:This part is missing information: Where the seabass used in the challenge Naï ve? haw were
they checked so that the author know for sure the seabass they used were not previously infected?
Response 16: Thanks for your advices. “120 Japanese seabass (body length, 6-8 cm; body weight, 7-9 g) passed the parasitological and microbiological examnination and were purchased from a farm in the Zhuhai city. A total of 120 healthy Japanese seabass were randomly divided into four groups with 30 fish per group and acclimated in tanks at 25 °C for seven days before the challenge test. Each group was established with three parallels.”
Point 17: The description of the preparation of the infected ZHLJ2019 is lacking information about the incubating equipment and procedure. The dilution solution used I guess was PBS but it is not
clear in the description. Where there replicates from the different treatment groups?
Response 17: Thanks for your advices. “oscillating incubator (ZHICHU, ZQTY-50N, Shanghai, China ) “ , “diluted to the corresponding concentration with PBS (0.01 M, pH 7.2).” “Each group was established with three parallels.”
Point 18: Line 143- The experiment ended six weeks after postinoculation. This is the same thing Chnge to- The experiment ended six weeks post-challenge.
Response 18: Thanks for your advice. I have resive “after postinoculation” to “post-challenge” .
Point 19: Line 157-159- The water temperature, salinity, and pH of the pond were 27 °C ± 2 °C, 6‰ ±
2‰, and 8.1 ± 0.1, respectively. The oxygen concentration was kept at 5.1 ± 1.5 mg/L.
Ammonia nitrogen and nitrite concentrations were 0.50 ± 0.10 and 0.60 ± 0.10 mg/L,
respectively. This is not results should be in the Martial and Methods
Response 19: Thanks for your advices. I have revise this sentence in the manuscript.(line 99-102)
Point 20: Disease characterization and clinical signs- should be rewritten as results it is currently half
M&M and half result. Start by describing the outbreak that morbid fish were collected from, year, and month the outbreak occurred as well as the mortality rates in one sentence then description of clinical signs and different analysis findings
Response 20: Thanks for your valuable advice. “In Septemtber 2019, we collected diseased Japanese seabass from a farm in three mud ponds in the Zhuhai City. Approximately 0.2%–0.5% mortality was recorded daily, peaking from September to November. The estimated rate of mortality of the Japanese seabass in this outbreak was approximately 30%. Diseased fish floated on the surface of water or was on the side of the pond, showing loss of appetite, lethargy, darkened body surface, and a small amount of bleeding spots on the body surface.”
Point 21: Morphological and biochemical characteristics- Need to present the similarities between
ZHLJ2019 and M. marinum shown in Table 2.
Response 21: Thanks for your advices. “The biochemical results of strain ZHLJ2019 were consistent with that of strain M. marinum ATCC927.”
Point 22: I would suggest a 16S sequencing comparison of the ZHLJ2019 strain and the M. marinum.
These can be easily done on BLAST. A full genome sequencing will be even better to make sure we are not talking about the same strain as described in other articles.
Response 22: Thanks for your advices. “such as M. marinum strain (ATCC 927) and M. basiliense with similarity exceeding 98%.”
Point 23:The results dialect and sentencing should be changed to a standard scientific journal dialect.
Ex. In the- Challenge experiment
Line 229-134- The first mortality occurred at 14 days postinjection, and moribund fish were characterized by diffuse erythema of the skin, raised scales, and swollen abdomens. The cumulative mortality of the control group and groups L (5 × 105 CFU/ml), M (5 × 106 CFU/ml), and H (5 × 107 CFU/ml) were 0, 75%, 90%, and 100%, respectively. Experimental results in Figure 5 showed the cumulative percent mortality in Japanese seabass exposed to M. marinum and mock-injected control fish over six weeks.
Response 23: Thanks for the your warm comment. I had revised this sentence in the manuscript.(line 268-272)
Point 24: I would also suggest to add statistics to the cumulative mortality.
Response 24: Thank your advices. I have added in the Figure 5.
Point 25: Line 255-257 To our knowledge, this study is the first report of mycobacteriosis caused by M. marinum infection in farmed Japanese seabass and extends the list of susceptible hosts of M. marinum. I would be cautious in stating a "This study is the first report of M. marinum infection in
farmed Japanese seabass and extends the list of susceptible hosts"
Response 25: Thank your advices. “This study is a new case of mycobacteriosis caused by M. marinum infection in farmed Japanese seabass.”
Point 26: Line 316-318 Thus, the structure of granulomas is closely related to the causative agent, host species, nutritional factors [35, 42], and their maturity [40]. The morphological variation of nodular lesions associated with different pathogens needs to be further studied.
The stricture of granulomas is closely related to different pathogens causing granulomas not
only Mycobacteria. The part the need to be investigated is in relation to different fish spp.,
organ and different Mycobacteria spp. I would clarify this part.
Response 26: Thanks your advices. I have revised “fish granuloma is caused by a variety of pathogens, and the morphological differences between different fish species and different pathogens are not clearly. Thus, the structure of granulomas is closely related to the causative agent, host species, nutritional factors [37, 44], and their maturity [42]. The morphological variation of nodular lesions associated with different fish spp., organs and different Mycobacteria spp. needs to be further studied.“
Point 27: The conclusion has to be related to what is presented or discussed in the paper.
The conclusion her talks about preventive measures, vaccines and zoonosis the conclusion
should be explained in relation to the findings.
Response 27: Thank your guidance. “This is the first case of mycobacteriosis among Janpanese seabass cultured in China. The causative agent of this case has been identified as M. marinum base on bacteriological, histopathological and molecular analysis. Rifampicin, isoniazid and erythromycin have antibacterial activity against the M. marinum strain. The results will lay the foundation to further research on diagnostic technology, virulence mechanisms, vaccine for preventing and controlling this disease in aquaculture and public health.”

Round 2
Reviewer 2 Report
Review 2:
Title: A Case of Mycobacteriosis in Cultured Japanese seabass (Lateolabrax japonicus) in Southern China
The corrected version is much better. Although, I still think this should be a case report, an important report, but it is not new, innovative or novel to publish it as a full article.
There are still some grammatical issues with this paper
Go through the paper and make sure that all spices of bacteria are in italic.
Just a few examples:
Abstract:
Line 17-18- In this study, the clinical signs and pathological characteristics of the disease, the pathogenicity and antibiotic sensitivity of pathogenic bacteria were investigated. To many The
Line 21-23- To insure that ZHLJ2019 isolate was the causative agent a Koch postulate trial was preformed
Line 26-M. marinum.The internal (Add space)
Line 28- wereconsistent with the classic pathological features of (Add space)
mycobacteriosis
Line 29- susceptibility of ZHLJ2019 isolate- Change to -to
Materials and Methods
2.6. Challenge experiment
Line 163-165- . 120 Japanese seabass (body length, 6-8 cm; body 163 weight, 7-9 g) passed the parasitological and microbiological examination and were purchased from a farm in the Zhuhai city.
Change this sentence to:
A hundred and twenty seabass (body length, 6-8 cm; body 163 weight, 7-9 g) were purchased from a farm in the Zhuhai city and went through a full parasitological and microbiological examination. The fish were found clean of any disease and there for were designated as healthy.
More: how many fish were examined from that group that was purchased from a farm in the Zhuhai city? What percent of fish were tested?
Line 167- Each group was established with three parallels. (How many fish per replicate? If there were 4 groups 30 fish per group that is 120 fish. you need to specify how many fish were per replication.
Use the term replicates not parallels.
Add what statistical program you used for the graphs
Results
Line 220- strain M. marinum-change to Italic
Line 270- Figure 270 5. Shows cumulative mortality- lower case letter
Line 274- (Fig. 4 274 K)were similar and Line 275- staining(Fig. 4 L). - Add space
A full genome sequencing will be even better to make sure we are not talking about the same strain as described in other articles.
Statistic should also include different letters describing the differences between the treatment groups not only statistical deviations.
Discussion
Lines 293-301- are in different font size then the rest of the paper.
Line 375- after receiving[47-48] and Line 377- osteomyelitis[49-50] – Add space
Lines 377-379- Humans may be infected when dealing with diseased fish, but there is a lack of drug sensitivity data (MIC) on M. marinum derived from diseased fish. Add reference.
Author Response
Dear Editor and Reviewers:
Thank you very much for your letter and the comments from the reviewers about our paper submitted to Fishes (Diagnoise and Treatment of Disease in Fish). The manuscript was carefully revised by using red text according to the comments. We responded point by point to each reviewer’s comments as listed below.
Changes for review issues are shown in blue. We supplemented “2.8 Statistical analysis”. Figure 5 was remade. Added some references.
Hope these will make it more acceptable for publication.
Thank you very much again.
Sincerely,
Zengchao, Huang
Point 1: Line 17-18- In this study, the clinical signs and pathological characteristics of the disease, the pathogenicity and antibiotic sensitivity of pathogenic bacteria were investigated. To many The
Response 1: Thanks for your advices. I have rewritten the sentence, “In this study, the clinical signs and pathological characteristics of diseased fish were investigated. Furthermore, the pathogenicity and antibiotic sensitivity of identified pathogenic bacteria from diseased fish were analyzed” (line17-19).
Point 2: Line 21-23- To insure that ZHLJ2019 isolate was the causative agent a Koch postulate trial was preformed
Response 2: Thanks for your advices. I have added this sentence in the maniscript (line 22-23).
Point 3: Line 26-M. marinum.The internal (Add space)
Response 3: Thanks for your advices. I have add space in “M. marinum. The ” (line 27).
Point 4: Line 28- wereconsistent with the classic pathological features of (Add space)
Response 4: Thanks for your advices. I have add space in “were consistent with”(line 29) .
Point 5: Line 29- susceptibility of ZHLJ2019 isolate- Change to -to
Response 5: Thanks for your advices. I have changed “of” into “to”(line 30).
Point 6: Line 163-165- . 120 Japanese seabass (body length, 6-8 cm; body 163 weight, 7-9 g) passed the parasitological and microbiological examination and were purchased from a farm in the Zhuhai city. Change this sentence to: A hundred and twenty seabass (body length, 6-8 cm; body 163 weight, 7-9 g) were purchased from a farm in the Zhuhai city and went through a full parasitological and microbiological examination. The fish were found clean of any disease and there for were designated as healthy
Response 6: Thanks for your advices. I have changed this sentence in the maniscript (line 164-166).
Point 7: More: how many fish were examined from that group that was purchased from a farm in the Zhuhai city? What percent of fish were tested?
Response 7: Thanks for your advices.The number of this batch of fish is about 200. Ten fish are randomly selected for testing, and the detection rate is about 5%. After passing parasitological and microbiological examnination, 120 fish were bought.
Point 8: Line 167- Each group was established with three parallels. (How many fish per replicate? If there were 4 groups 30 fish per group that is 120 fish. you need to specify how many fish were per replication.
Response 8: Thanks for your advices. Each replicate has ten fish (line 170).
Point 9: Use the term replicates not parallels.
Response 9: Thanks for your advices. I have changed “parallels” into “replicates” (line 170).
Point 10: Add what statistical program you used for the graphs
Response 10: Thanks for your advices. Figure 5 was construct by Graphpad Prism 8.0.1. (line 183).
Point 11: Line 220- strain M. marinum-change to Italic
Response 11: Thanks for your advices. I have changed “M. marinum” into “M. marinum”(line 229).
Point 12: Line 270- Figure 270 5. Shows cumulative mortality- lower case letter
Response 12: Thanks for your advices. I have changed this error (line 279).
Point 13: Line 274- (Fig. 4 274 K)were similar and Line 275- staining(Fig. 4 L). - Add space
Response 13: Thanks for your advices. I have added space in “(Fig. 4 K) were” and “staining (Fig. 4 L)” (line 283-284).
Point 14: A full genome sequencing will be even better to make sure we are not talking about the same strain as described in other articles
Response 14: Thanks for your advices. Due to time constrants, we did not conduct a full genome sequencing in this study, and we will conduct a full genome sequencing in the following research.
Point 15: Statistic should also include different letters describing the differences between the treatment groups not only statistical deviations.
Response 15: Thanks for your advices. I had added the different describing the differences between the treatment groups in Figure 5 and described in “2.8 Statistical analysis” (line 195-201) .
Point 16: Lines 293-301- are in different font size then the rest of the paper.
Response 16: Thanks for your advices. I have revised the error (line 302-311).
Point 17: Line 375- after receiving[47-48] and Line 377- osteomyelitis[49-50] – Add space
Response 17: Thanks for your advices. I have added space in “receiving [49-50] ”and “osteomyelitis [51-52]”
(line 385-387)
Point 18: Lines 377-379- Humans may be infected when dealing with diseased fish, but there is a lack of drug sensitivity data (MIC) on M. marinum derived from diseased fish. Add reference.
Response 18: Thanks for your advices.I have added reference. “Humans may be infected when dealing with diseased fish, but there is a lack of drug sensitivity data (MIC) on M. marinum derived from diseased fish [53-55].”(line 389).
